# Hippocampal Leptin Resistance and Cognitive Decline: Mechanisms, Therapeutic Strategies and Clinical Implications

**DOI:** 10.3390/biomedicines12112422

**Published:** 2024-10-22

**Authors:** Ismael Valladolid-Acebes

**Affiliations:** The Rolf Luft Research Center for Diabetes and Endocrinology, Karolinska Institutet, Karolinska University Hospital L1, SE-171 76 Stockholm, Sweden; ismael.valladolid.acebes@ki.se; Tel.: +46-72-873-56-52

**Keywords:** leptin resistance, obesity, hypothalamus, hippocampus, synaptic plasticity, spatial learning, memory, cognitive decline, leptin sensitizers, clinical trials

## Abstract

**Background:** Leptin, an adipokine essential for regulating energy balance, exerts important effects on brain function, notably within the hippocampus, a region integral to learning and memory. Leptin resistance, characterized by diminished responsiveness to elevated leptin levels, disrupts hippocampal function and exacerbates both obesity and cognitive impairments. **Scope:** This review critically examines how leptin resistance impairs hippocampal synaptic plasticity processes, specifically affecting long-term potentiation (LTP) and long-term depression (LTD), which are crucial for cognitive performance. **Findings:** Recent research highlights that leptin resistance disrupts N-methyl-D-aspartate (NMDA) receptor dynamics and hippocampal structure, leading to deficits in spatial learning and memory. Additionally, high-fat diets (HFDs), which contribute to leptin resistance, further deteriorate hippocampal function. Potential therapeutic strategies, including leptin sensitizers, show promise in mitigating brain disorders associated with leptin resistance. Complementary interventions such as caloric restriction and physical exercise also enhance leptin sensitivity and offer potential benefits to alleviating cognitive impairments. **Aims of the review:** This review synthesizes recent findings on the molecular pathways underlying leptin resistance and its impact on synaptic transmission and plasticity in the hippocampus. By identifying potential therapeutic targets, this work aims to provide an integrated approach for addressing cognitive deficits in obesity, ultimately improving the quality of life for affected individuals.

## 1. Introduction

Leptin, a 16 kDa polypeptide hormone encoded by the *ob* gene, is a key regulator of energy homeostasis with profound implications for metabolic functions [1,2]. First described in 1994 [1], the role of leptin extends beyond its foundational action in regulating appetite and energy expenditure. It acts as a critical neuroendocrine signal, influencing a broad spectrum of physiological processes through its interaction with an array of receptors distributed across nearly every tissue in the body [3,4,5,6,7,8,9,10,11].

The biological actions of leptin are linked to the leptin receptor (ObR), which manifests in multiple isoforms with varying functions [4,11,12,13,14]. The long isoform, or ObRb, is predominantly expressed in the hypothalamus and is crucial for mediating the anorexigenic effects of the hormone and integrating metabolic signals to influence food intake and energy expenditure [13,14,15,16,17]. Conversely, soluble isoforms like ObRe modulate leptin levels in the circulation, thereby buffering against excessive hormonal fluctuations [11,18,19,20,21,22]. The complex signaling cascades initiated by leptin binding—particularly through the Janus kinase/signal transducer and activator of transcription proteins (Jak/STATs) pathway, alongside the phosphoinositide 3-kinases (PI3Ks) and mitogen-activated protein kinase (MAPK) pathways—underscore its role in cellular processes such as metabolic regulation, synaptic plasticity and neuronal survival [4,6,23,24,25,26,27,28].

However, in the context of obesity and metabolic disorders, the physiological actions of leptin are compromised by the development of leptin resistance [2,29]. This condition, characterized by impaired leptin signaling despite elevated circulating levels, contributes to a paradoxical increase in body weight and metabolic dysregulation [29,30,31,32,33]. The persistence of leptin resistance in obesity and its impact on cognitive function, particularly within the hippocampus, highlights a critical intersection between metabolic and neurological health [34,35] (Figure 1).

The hippocampus, located in the medial temporal lobe, is vital for both spatial and declarative memory [36]. It processes information through a complex network of neurons organized in a distinct laminar structure, receiving input from various pathways, such as the perforant path, mossy fibers and Schaffer collaterals [36,37,38,39]. This structure is essential for encoding spatial information and consolidating short- and long-term memories [36]. Neuronal plasticity within the hippocampus, facilitated by mechanisms like long-term potentiation (LTP) and long-term depression (LTD), underpins learning and memory processes both in rodents and in humans [40].

Preclinical evidence suggests that leptin resistance impairs hippocampal synaptic plasticity and cognitive performance, exacerbating learning and memory deficits [41,42,43,44,45,46]. The effects of high-fat diets (HFDs) and other metabolic perturbations on leptin signaling in the mouse brain point to a broader impact on cognitive function and suggest potential therapeutic avenues (Figure 1). In-field research demonstrates the therapeutic potential of leptin analogs and sensitizers [47,48,49], which can be potentially beneficial for cognitive deficits in both rodents and humans, albeit complicated by the phenomenon of leptin resistance.

Despite significant advances in understanding the relationship between hippocampal leptin resistance and cognitive decline, several gaps in knowledge remain. Firstly, the exact molecular mechanisms linking leptin resistance to synaptic plasticity and cognition are unclear [40,41,42,43,44,45,46]. Additionally, the impact of leptin resistance at different life stages and its role in hippocampal function are poorly understood [46]. Finally, translational challenges from animal models to humans persist, and while therapies like leptin sensitizers show promise [47,48,49], their effectiveness in treating cognitive decline needs further research for clinical application.

This review seeks to offer a comprehensive examination of the influence of leptin resistance on hippocampal function, with particular emphasis on the mechanisms affecting glutamatergic transmission and synaptic plasticity in the context of obesity. By synthesizing recent and past findings on the molecular pathways involved in leptin resistance and their implications for cognitive function, this review aims to identify potential therapeutic targets to address cognitive deficits linked to leptin resistance in obesity.

## 2. Leptin: Biological Role and Actions

Leptin is a 167-amino acid polypeptide encoded by the *ob* gene [1]. It starts with a 2-amino acid signal sequence that is cleaved off before the protein enters the bloodstream. Consequently, the mature leptin molecule, with a molecular weight of 16 kDa, consists of 146 amino acids and resembles class I cytokines in its tertiary structure [1,50].

The biological activity of leptin hinges on a critical disulfide bond in its structure. Despite some interspecies variation, human leptin is a highly conserved protein; for example, sharing a homology of 84% with mouse leptin and 83% with rat leptin [1,2,51,52,53]. While predominantly produced in white adipose tissue [1,2,51,52,53,54,55], leptin is also synthesized by brown adipose tissue and various other organs, including the brain, skeletal muscles, stomach, breasts, ovaries and placenta (during pregnancy) [56,57,58,59,60,61,62,63,64,65].

Leptin synthesis is influenced by several factors. For instance, age, HFDs, insulin and glucocorticoids all enhance leptin production, while fasting, exercise, heat and adrenergic stimulation suppress it [62,66,67]. Its secretion follows a pulsatile pattern and is modulated by other hormones like insulin and cholecystokinin [67]. Additionally, leptin secretion exhibits a circadian rhythm, with peak levels occurring during the day in humans and at night in rodents [68,69,70].

Once in circulation, leptin can either be free or bound to plasma proteins [68]. Its plasma concentrations vary from around 5 ng/mL in individuals with a normal weight and can reach up to 25–100 ng/mL in those with obesity [2]. The hormone has a short half-life of 25 min in its endogenous form and 90 min in its exogenous form, with rapid clearance primarily through the kidneys [71].

Leptin is a key player in the neuroendocrine and behavioral regulation of food intake and energy balance [2,51,52,53,54,55]. It has an extensive range of effects, both centrally in the brain and peripherally across various tissues. This wide-ranging influence is largely attributed to its receptors, which are distributed throughout nearly every tissue in the body.

## 3. Leptin Receptors

The leptin receptor, first discovered in 1995, is a membrane-bound class I cytokine receptor [12]. Researchers have identified up to six isoforms of the leptin receptor: ObRa, ObRb, ObRc, ObRd, ObRe and ObRf. These isoforms all feature an N-terminal extracellular domain for leptin binding, a transmembrane domain of 34 amino acids anchoring the receptor to the cell membrane and a variable C-terminal intracellular domain [4,13,14]. The diversity among these isoforms leads to their classification into three categories: short isoforms (ObRa, ObRc, ObRd, ObRf) with 30–40 amino acids in the C-terminal region; the long isoform (ObRb) with a 302-amino acid C-terminal; and the soluble isoform (ObRe), which lacks this domain altogether [12,13,14].

The soluble isoform, ObRe, plays a crucial role in modulating leptin levels in the bloodstream by binding to circulating leptin and acting as a buffer [16,17,18,19,20]. In contrast, the long isoform ObRb, highly expressed in the hypothalamus, is central to the anorexigenic effects of leptin [12,13,14,15]. ObRb uniquely contains the intracellular domain required for leptin signal transduction, primarily through the Jak/STAT pathway [4,13,14,15,16,23].

Short isoforms of the leptin receptor are involved in recruiting Jak proteins and appear to play roles in leptin internalization, degradation and transport across the blood–brain barrier and the choroid plexus [13,24,26,27,28] (Figure 1). Leptin receptors are widely distributed throughout the body, with expression detected in various regions of the central nervous system (CNS), including the hypothalamus, cerebral cortex, ventral tegmental area, nucleus of the solitary tract and hippocampus [3,4,5,7,8,9,10]. Additionally, they are found in peripheral tissues such as the spleen, liver, skeletal muscle, heart and adipose tissue [6]. This widespread distribution highlights the pivotal role of leptin in regulating numerous physiological processes throughout the body.

### 3.1. Intracellular Signaling of Leptin Coupled to the Leptin Receptor

The Jak/STAT pathway is the primary intracellular signaling cascade of leptin coupled to ObRb. However, ObRb and, to a lesser extent, ObRa can also activate alternative pathways, such as the PI3K and the MAPK pathways [4,6,23,24,25,26,27,28] (Figure 2).

#### 3.1.1. Jak/STAT Pathway

In the hypothalamus, the leptin-driven activation of STAT3 is closely linked to weight loss and increased energy expenditure in physiological conditions [2,17,23,52,66,72,73] (Figure 2). Activated or phosphorylated STAT3 acts as a pivotal transcription factor, orchestrating the expression of genes like suppressor of cytokine signaling 3 (*SOCS3*) and the tyrosine-protein phosphatase non-receptor type 1B (*PTP1b*), both of which in turn mediate crucial feedback inhibition in the leptin receptor [2,74] (Figure 2). Beyond the hypothalamus, in areas like the hippocampus, leptin influences neuronal survival through the Jak/STAT and PI3K pathways by enhancing the production of antioxidants, such as manganese superoxide dismutase and the anti-apoptotic protein Bcl-xL (B-cell lymphoma-extra-large) [75]. Additionally, leptin modulates hippocampal neuronal excitability by increasing intracellular Ca^2+^ concentrations in glutamatergic neurons induced by N-methyl-D-aspartate (NMDA) receptor facilitation [76] (Figure 2). This results in modifications to AMPA receptor trafficking, which in turn induces persistent changes in the excitatory synaptic strength within the hippocampus [77] (Figure 2). Rodents deficient in leptin signaling or receptors exhibit notable deficits in spatial learning and memory, highlighting the role of leptin in hippocampal synaptic plasticity and cognitive function [41,78,79,80].

#### 3.1.2. PI3K/Akt Pathway

The PI3K pathway, shared by leptin and insulin signaling, is activated when leptin binds to its receptor, leading to the phosphorylation of insulin receptor substrates (IRS1/2) and the subsequent activation of protein kinase B (PKB/Akt) [6,25,26,27,28] (Figure 2). Akt regulates numerous physiological processes, including glucose metabolism, cell proliferation, apoptosis, transcription and migration [27,28]. In the hippocampus, Akt activation is crucial for synaptic plasticity, with studies showing that the PI3K/Akt pathway governs LTD through NMDA receptor synthesis in the CA1 region [81,82,83,84,85]. This underlines the need of a fully functional PI3K/Akt pathway for leptin-mediated synaptic plasticity within the hippocampus.

#### 3.1.3. MAPK Pathway

Leptin also activates the MAPK pathway via its receptors [4,13,14]. Leptin binding prompts the phosphorylation of Tyr985, which facilitates the binding of extracellular signal-regulated kinases 1/2 (ERK1/2) and the activation of transcription factors like c-Fos and Erg-1 (early growth response protein 1), involved in cell proliferation and differentiation [27,28] (Figure 2). In the hippocampus, MAPK activation counters the effects of the PI3K/Akt pathway. Electrophysiological studies show that inhibiting MAPK prevents increased neuronal excitability [85,86,87,88], and in vitro research suggests complementary mechanisms between PI3K/Akt and MAPK pathways in regulating synaptic plasticity [89]. Notably, peak Erg-1 induction correlates with high MAPK activity and low phosphorylated Akt levels, indicating that the interplay between these pathways is crucial for hippocampal synaptic plasticity and cognitive processes [86,87,88,89,90,91].

## 4. Physiological Effects of Leptin

Leptin orchestrates both food intake and energy expenditure through complex central and peripheral mechanisms. In the hypothalamus, leptin exerts its effects by modulating the expression of neuropeptides (Figure 2). It is known that leptin decreases orexigenic peptides, like neuropeptide Y (NPY) and Agouti-related peptide (AgRP), while increasing anorexigenic peptides, such as pro-opiomelanocortin (POMC) and cocaine- and amphetamine-regulated transcript (CART) [29,92,93,94,95,96] (Figure 2). Additionally, leptin enhances neurotrophin expression, including brain-derived neurotrophic factor (*BDNF*), in the ventromedial nucleus [97,98,99]. Disruptions in these pathways, along with hyperleptinemia and central leptin resistance, may contribute to reductions in neurogenesis, altered neuronal plasticity and cognitive deficits [97,98,99,100] (Figure 1).

Peripherally, leptin boosts energy expenditure and stimulates fatty acid β-oxidation in the mouse liver [101,102]. In adipose tissue, leptin administration promotes lipolysis in ob/ob mice but not in db/db mice lacking the leptin receptor [103]. In rat skeletal muscles, leptin enhances β-oxidation via AMPK activation and improves insulin sensitivity by reducing triglyceride accumulation [104]. Cardiac tissue from rats also benefits from leptin, which stimulates β-oxidation independently of AMPK and acts as a cardiotrophic factor, promoting cardiomyocyte growth [105,106].

Over the years, research has increasingly focused on the actions of leptin on neuronal functions beyond energy regulation [36]. Preclinical suggests that leptin influences excitatory synaptic transmission and plasticity in the hippocampus [36,76,80,85,87,88,107] (Figure 1 and Figure 2). Direct leptin administration in the dentate gyrus or the CA1 region of the hippocampus enhances LTP and spatial memory in rodents [108,109]. Moreover, acute leptin application in hippocampal slices facilitates NMDA receptor-dependent synaptic plasticity, converting short-term potentiation into LTP and enhancing its induction [76,110] (Figure 2). Leptin also induces depotentiation, a novel form of LTD dependent on NMDA receptor synthesis, which counters hyperexcitability and maintains synaptic balance [82,83] (Figure 2). Furthermore, leptin increases dendritic spine density and mobility, highlighting its role in regulating excitatory synaptic efficacy [111].

In clinically relevant disease models, dysfunctional leptin signaling has been implicated in a number of hippocampal impairments. For instance, genetically engineered mouse models lacking leptin signaling exhibit impaired hippocampal plasticity, including altered LTP and LTD, that correlates with spatial learning and memory deficits [41,79,112]. In murine models of diet-induced obesity (DIO), impaired hippocampal leptin signaling has been shown to compromise spatial learning and memory and disrupt hippocampal glutamate metabolism and neurochemistry [113,114]. Moreover, aberrant leptin signaling adversely affects synaptic plasticity within the hippocampus, creating alterations in LTD as well as changes in dendritic density and morphology [34,35,113,114]. Importantly, the detrimental impact of impaired leptin signaling on hippocampal function and cognitive performance observed in murine models parallels human data, thus validating the impact of dietary patterns and fat intake on brain morphology and cognitive health across the lifespan [115,116,117,118].

## 5. Hippocampal Leptin Resistance

Leptin resistance represents a pivotal pathophysiological aspect of obesity, characterized by an impairment in the ability of the CNS to respond to leptin [29] (Figure 1). Leptin typically functions to inhibit food intake and enhance energy expenditure, thereby facilitating weight loss [90,91,92,93,94] (Figure 2). However, in the state of leptin resistance, the response of the brain to leptin is compromised despite elevated circulating levels of the hormone, leading to an unremitting increase in body weight [2,29] (Figure 1).

In the context of obesity, an excessive intake of calories results in adipocyte hypertrophy and hyperplasia (Figure 1), which precipitates a state of chronic low-grade inflammation and the subsequent infiltration of immune cells within the adipose tissue [119,120,121]. This inflammatory milieu is associated with an augmented release of pro-inflammatory cytokines and a perturbation in the homeostasis of crucial adipokines, including leptin and adiponectin [119,120,121]. Empirical evidence derived from obesity models indicates that, despite elevated leptin concentrations, the anticipated physiological responses are not achieved due to the underlying resistance mechanisms [2,29] (Figure 1).

The molecular mechanisms of leptin resistance involve perturbations in critical signaling pathways, such as STAT3, PTP1b and SOCS3, as well as alterations in the leptin receptor [2,29,33,122,123,124,125,126]. These disruptions impede the effective transduction of leptin signals, thereby exacerbating the obesity phenotype.

Leptin resistance has major implications for hippocampal function and plasticity [41,79,112,114,115,116,117,118] (Figure 1). In physiological conditions, leptin facilitates synaptic plasticity by inducing LTP and modulating glutamatergic transmission [108,109] (Figure 2). However, in leptin-resistant conditions, such as those resulting from HFDs and age-related impairments, these actions are diminished or absent [34,35,113,114,115,116,117,118] (Figure 1). For example, the ability of leptin to induce LTP or promote LTD at CA1 synapses is reduced, if not completely lost, in the hippocampus of DIO and aged mice [114,115,116,117,118].

In animal models of leptin resistance, deficits in hippocampal-dependent behaviors are observed [41,42,43,44,45,46]. For instance, leptin-resistant rodents exhibit impaired spatial learning and memory in water maze tasks, despite normal locomotor activity [41,127]. Studies of genetic models like the Zucker rats demonstrate important cognitive impairments, particularly under conditions that challenge hippocampal plasticity, such as tasks requiring longer inter-trial intervals [42].

These changes suggest that leptin resistance disrupts normal hippocampal function by impairing synaptic plasticity, which underlies cognitive deficits [41,42,43,44,45,46] (Figure 1). Furthermore, leptin resistance often accompanies other metabolic disturbances that may compound these effects [41,42,43,44,45,46]. Thus, addressing leptin resistance and its impact on hippocampal function is crucial for understanding and mitigating cognitive impairments associated with metabolic disorders and aging.

## 6. Cognitive Consequences of Hippocampal Leptin Resistance

The consequences of hippocampal leptin resistance extend beyond metabolic dysregulation to encompass profound effects on learning, memory and reward processing, which in turn can exacerbate overeating and obesity in rodents and humans [36,41,42,43,44,45,46,128]. Emerging evidence shows that hippocampus-dependent mechanisms play a crucial role in the regulation of food intake by influencing higher-order cognitive processes that govern decision-making and impulse control [46,98,129,130,131,132]. The hippocampus, traditionally recognized for its role in learning and memory, also modulates energy homeostasis and eating behaviors, suggesting a critical intersection between cognitive function and metabolic control [46,98,129,130,131,132]. The disruption of these processes through DIO, particularly via the consumption of HFDs and palatable foods, not only leads to hippocampal dysfunction but also triggers addictive-like behaviors that reinforce overeating [133,134].

Research has shown that high-calorie diets adversely affect hippocampal structure and function, impairing synaptic plasticity and neurogenesis, which are essential for learning and memory [78,98,135,136,137,138]. This suggests that the hippocampus is highly sensitive to dietary composition, particularly during critical developmental periods when the brain is more vulnerable to external insults, including neurotoxic agents like saturated fats [46,139]. The link between DIO and cognitive decline, including memory loss and an increased risk of neurodegenerative diseases, points out the potential long-term consequences of poor dietary habits established during adolescence, a period marked by substantial synaptic pruning and shifts in glutamatergic transmission [114,140,141,142].

Particularly concerning is the impact of HFDs on NMDA receptor function, which plays a pivotal role in synaptic plasticity and cognitive function. Developmental changes in NMDA receptor subunits, notably the transition from NR2B to NR2A, are crucial for synaptogenesis and neuronal connectivity, yet the effects of HFDs on these receptors during adolescence, as compared to adulthood, remain underexplored [143,144,145,146]. Understanding these mechanisms is essential, as the NMDA receptor has been proposed as a therapeutic target for treating eating disorders associated with obesity [147,148,149]. The potential irreversibility of obesity-induced hippocampal damage highlights the urgency of early interventions to prevent cognitive impairments and to address the addictive nature of food that undermines efforts to control intake, contributing to the perpetuation of obesity [150].

## 7. Implications of Hippocampal Leptin Resistance for Executive Function and Alzheimer’s Disease

In rodent models and humans, obesity-induced leptin resistance in the hippocampus has significant implications for executive functions, including decision-making, impulse control and attention regulation, which are vital for maintaining cognitive control over food intake [131,151,152,153,154,155]. When leptin resistance occurs in the hippocampus, these cognitive processes are impaired, leading to weakened appetite regulation and reinforcing maladaptive eating behaviors [131,151,152,153,154,155]. This dysfunction can diminish working memory and executive function, making it challenging for individuals to prioritize long-term goals over immediate food-related impulses [131,151,152,153,154,155]. These derangements become even more pronounced in neurodegenerative conditions such as Alzheimer’s disease [85,116,156,157]. In patients with Alzheimer’s disease, leptin signaling in the hippocampus is decreased and leptin localization is shifted, being more abundant in reactive astrocytes and less reactive in neurons [158]. This deranged hippocampal leptin signaling can very well be a contributing factor to neuronal degeneration, linking metabolic dysfunction directly to cognitive decline in Alzheimer’s disease [158].

Both leptin resistance in obesity and disrupted leptin signaling in Alzheimer’s disease share a common pathway of cognitive impairment, particularly affecting memory and executive function both in rodents and humans [85,116,156,157,158]. This relationship underscores the potential for targeting leptin pathways as a therapeutic strategy to mitigate cognitive decline and address the metabolic aspects of neurodegenerative diseases [85,116,156,157,158]. By understanding these mechanisms, we can better explore interventions that may enhance leptin sensitivity and improve cognitive outcomes in affected individuals.

## 8. Therapeutic Interventions and Future Directions

Emerging evidence suggests that targeting leptin signaling could be a promising therapeutic strategy to mitigate the cognitive decline associated with hippocampal leptin resistance [159,160,161]. The therapeutic potential of leptin has been proven by studies demonstrating that supplementation with the hormone can reverse cognitive deficits in leptin-deficient models, in addition to its broader neurotrophic effects in regions such as the cerebellum and anterior cingulate gyrus [162,163,164].

In individuals with leptin deficiency, leptin replacement therapy not only normalizes metabolic abnormalities but also induces structural and functional improvements in key areas involved in cognitive and emotional processes, including the hippocampus [162,163,164,165,166]. Importantly, it has been shown that the exogenous administration of leptin improves neurogenesis, axonal growth, synaptogenesis, LTP and memory formation [77,167,168,169]. Therefore, restoring leptin signaling in the brain could be a viable approach to prevent or even reverse hippocampal atrophy and cognitive decline in patients with leptin deficiency.

However, the clinical application of leptin therapy presents significant challenges, particularly due to the phenomenon of leptin resistance in obesity [29]. In leptin-resistant individuals, elevated leptin levels fail to exert their normal physiological effects, reducing the efficacy of leptin therapy [29]. This is analogous to insulin resistance in type 2 diabetes, where high insulin levels do not produce the expected metabolic effects [170]. In this context, the development of leptin sensitizers, such as amylin analogs and Celastrol, which have shown promise in preclinical models [47,48,49], may offer a novel therapeutic avenue.

Recent clinical trials have explored various amylin analogs, including pramlintide, cagrilintide, davalintide (AC2307) and GUBamy (GUB014295), that have demonstrated potential in addressing overweight and/or obesity due to their actions as leptin sensitizers [47,171] (Table 1). These analogs might represent an important therapeutic venue in the future to overcome brain derangements associated with leptin resistance [172,173,174]. For instance, trials like NCT00690235, which focused on schizophrenia and medication-induced weight gain, and NCT03560960, which was aimed at Alzheimer’s disease and mild cognitive impairment, highlight the broader implications of improving leptin sensitivity in the context of brain dysfunction (Table 1). Thus, pramlintide appears to be a promising candidate for not only facilitating weight loss and enhancing glycemic control but also potentially offering cognitive benefits through improved metabolic function [172,173].

Similarly, cagrilintide, a novel amylin analog, has proven its safety and efficacy at reducing weight and improving metabolic outcomes in patients with overweight or obesity [175,176,177,178,179], and has a number of studies in phase III (Table 1). The relevance of these outcomes might extend to neurological conditions, where metabolic health plays a critical role in cognitive function [47,160,161]. For example, the potential role of cagrilintide in improving insulin sensitivity, reducing cardiovascular risk markers and regulating energy expenditure may offer therapeutic benefits in brain disorders exacerbated by metabolic impairments, such as Alzheimer’s disease [47,175,176,177,178,179].

In addition to amylin analogs, Celastrol, a plant-derived compound, has been investigated for its potential as a leptin sensitizer (Table 1). A clinical trial (NCT05494112) is currently underway to evaluate the efficacy of Celastrol in treating obesity by restoring leptin sensitivity [48] (Table 1), which may have important implications for neurodegenerative diseases in the future.

Preclinical studies suggest Celastrol enhances leptin signaling, potentially reversing leptin resistance, promoting weight loss and improving brain function in metabolic impairment contexts [48]. Although clinical data is still emerging, Celastrol represents a promising approach to overcoming the limitations of leptin therapy in resistant individuals [48], with potential benefits for both metabolic and cognitive health.

Collectively, these clinical trials highlight an increasing interest in utilizing leptin sensitizers to tackle not only obesity and metabolic disorders but also neurological conditions. This provides renewed hope for individuals suffering from cognitive impairments associated with leptin resistance and metabolic dysfunction.

In addition to pharmacological approaches, lifestyle interventions that modulate leptin sensitivity, such as caloric restriction and physical exercise, merit further investigation [180,181,182]. These interventions have been shown to enhance leptin signaling pathways [183,184] and may serve as adjunctive therapies to leptin sensitizers or replacement therapy.

It should, however, be acknowledged that the pleiotropic effects of leptin on neurotrophic factors, insulin and sex hormones complicate the interpretation of its direct effects on brain structure and function [26,185,186,187]. 

Future research should also focus on disentangling these complex interactions to better understand how leptin influences the hippocampus independently of its metabolic actions. This understanding is essential for developing targeted interventions that enhance leptin-dependent hippocampal function without exacerbating peripheral resistance to the hormone.

In summary, while the role of leptin in cognitive function is increasingly recognized, translating these findings into effective clinical therapies will require a nuanced approach. Addressing leptin resistance and optimizing leptin delivery to the brain represent critical future directions in the effort to combat cognitive decline and hippocampal dysfunction. Collaborative efforts integrating clinical trials, basic research and novel drug development will be essential to fully harness the therapeutic potential of leptin, its analogs and its sensitizers in neurodegenerative diseases.

## 9. Clinical Implications and Outlook

Hippocampal leptin resistance has relevant clinical implications for cognitive decline, particularly within the context of obesity and metabolic disorders [47,116,169,188] (Figure 1). The hippocampus, a critical region for learning, memory and decision-making, suffers when leptin signaling is disrupted, resulting in compromised synaptic plasticity and neurogenesis [34,36,44,98,131] (Figure 1). Such disruptions contribute to the cognitive deficits, including memory impairment and compromised spatial learning, frequently observed in individuals with leptin resistance [116,131,169,188,189] (Figure 1). Additionally, leptin resistance intensifies addictive-like behaviors associated with overeating, establishing a detrimental cycle that exacerbates obesity and further impairs hippocampal function [41,42,43,45,46,47,48,50,184,185,190,191]. Clinically, these findings underscore the unmet need for early intervention strategies targeting leptin resistance.

Potential approaches include pharmacological agents designed to enhance leptin sensitivity, such as amylin analogs or Celastrol, as well as lifestyle modifications like caloric restriction and increased physical activity [47,48,171,172,173,174,175,176,177,178,179,180,181,182,183,184] (Table 1). Addressing leptin resistance in the hippocampus may not only help mitigate cognitive decline but could also alleviate obesity-related comorbidities, including the risk of neurodegenerative diseases [188]. However, the complexity of the role of leptin in the CNS presents challenges [4,6,23,24,25,26,27], requiring further research to refine therapeutic strategies that enhance leptin signaling specifically within the brain without worsening peripheral leptin resistance. This emphasizes the critical importance of understanding and addressing hippocampal leptin resistance in the development of targeted, multifaceted treatments for cognitive health.

## 10. Conclusions

In conclusion, emerging understanding of hippocampal leptin resistance has revealed its profound impact on cognitive function [36,41,42,43,44,45,46,98,114,128,129,130,131,132,139,140,141,142,143,144,145,146,147,148,149,150], particularly in executive processes, such as decision-making, impulse control and memory [131,151,152,153,154,155]. This dysregulation not only exacerbates the challenges of obesity but also contributes to the pathophysiology of neurodegenerative diseases like Alzheimer’s disease [85,116,156,157]. As leptin signaling is compromised in these conditions, the link between metabolic dysfunction and cognitive decline becomes increasingly apparent [158].

Addressing hippocampal leptin resistance presents a promising avenue for therapeutic intervention [159,160,161]. Targeted strategies, including the development of leptin sensitizers and lifestyle modifications [47,48,171,172,173,174,175,176,177,178,179,180,181,182,183,184], may enhance cognitive outcomes while also mitigating the obesity epidemic [192]. These approaches hold the potential to restore synaptic plasticity and improve neurogenesis, critical components for maintaining cognitive health (Table 1, NCT03560960).

However, the complexity of the role of leptin in the central nervous system needs further investigation into the mechanisms underlying its effects [4,6,23,24,25,26,27]. Unraveling these pathways could lead to innovative treatments that specifically enhance leptin signaling within the brain, thereby alleviating cognitive deficits without exacerbating peripheral resistance [85,116,156,157,158].

Given the increasing prevalence of obesity and its associated cognitive impairments [156,192], early interventions targeting leptin resistance are essential. Ultimately, fostering an integrated understanding of the interplay between metabolism and cognition [158] will be crucial in developing effective strategies to improve brain health and quality of life for individuals affected by these conditions. The time is ripe for a multifaceted approach that bridges clinical research, pharmacotherapy and lifestyle interventions [47,48,171,172,173,174,175,176,177,178,179,180,181,182,183,184] to combat the deleterious effects of hippocampal leptin resistance and enhance cognitive resilience.

## Figures and Tables

**Figure 1 biomedicines-12-02422-f001:**
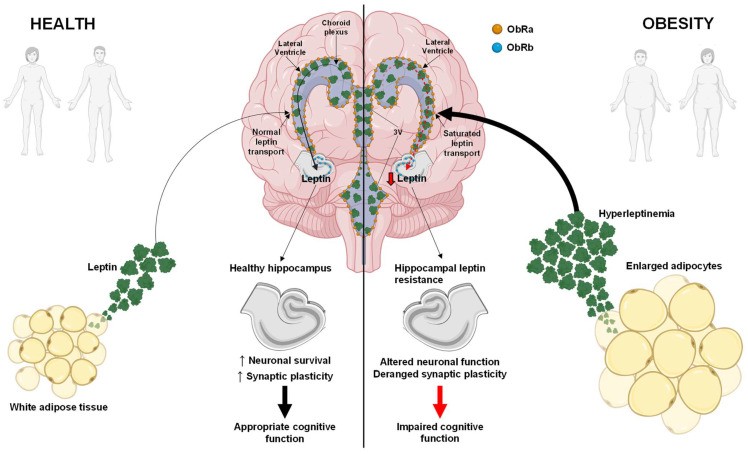
Synoptic overview of the hippocampal regulatory actions of leptin in both health and disease. In a healthy state (**left**), leptin is secreted from adipocytes at a physiological level and easily crosses the blood–brain barrier through an ObRa-dependent saturable transport process to exert its effects on the brain [24,28]. Within the hippocampus, leptin promotes cognitive function by swiftly modulating neuronal survival and synaptic plasticity. However, in obesity (**right**), leptin levels are elevated, resulting in leptin resistance and diminished transport into the brain. This impaired leptin signaling in the hippocampus is linked to disrupted neuronal function and synaptic plasticity, thus increasing the risk of cognitive decline. Created with Biorender.com.

**Figure 2 biomedicines-12-02422-f002:**
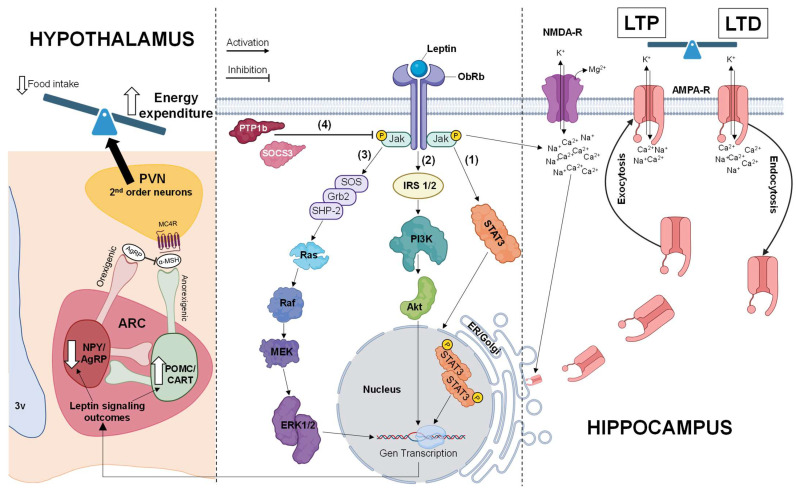
Physiological leptin signal transduction and its effects in the hypothalamus and hippocampus. Upon leptin binding to ObRb (**middle**), Jak2 undergoes phosphorylation, triggering the dimerization and nuclear translocation of STAT3 (1). Concurrently, the PI3K/Akt (2) and MAPK (3) pathways are activated following Jak2 phosphorylation, resulting in nuclear gene transcription. ObRb signaling is negatively regulated by the activation of SOCS3 and PTP1B (4). Leptin receptor signaling in the hypothalamus (**left**) stimulates the anorexigenic POMC/CART neurons. The release and further cleavage of POMC yields the melanocortin-stimulating hormone (α-MSH), which signals through melanocortin receptors, leading to anorexigenic effects. In contrast, leptin signal transduction represses the expression of the orexigenic neuropeptides NPY/AgRP in neighboring neurons. AgRp also acts as an inverse agonist of melanocortin receptors, counteracting the effects of α-MSH. To regulate energy homeostasis, these first-order neuronal targets of adiposity signals project from the arcuate nucleus (ARC) to the paraventricular nucleus (PVN) and other regions of the brain. As a consequence, food intake is reduced in favor of increased energy expenditure. In the hippocampus (**right**), activation of hippocampal ObRb enhances NMDA receptor (NMDA-R) function, leading to modifications in AMPA receptor (AMPA-R) trafficking, which subsequently promotes sustained changes in excitatory synaptic strength. Abbreviations: ERK1/2, extracellular signal-regulated kinase 1/2; ER, endoplasmic reticulum; Grb2, growth factor receptor-bound protein 2; MCR-4, melanocortin receptor 4; MEK, mitogen-activated protein kinase; Raf, proto-oncogene serine/threonine-protein kinase; Ras, rat sarcoma virus small GTPase; SHP-2, Src homology region 2 domain-containing phosphatase-2; SOS, Son of Sevenless guanine nucleotide exchange factor. Created with Biorender.com.

**Table 1 biomedicines-12-02422-t001:** Ongoing clinical trials in obesity and/or overweight using amylin analogs or Celastrol (up to 16 October 2024).

	NCT Number Phase/Design	Drug Name(s)	Condition(s)	Primary Outcome(s)	Sponsor	Status	Last Updated Posted
	Pramlintide						
1	NCT03560960 Early Phase I/Diagnostic	Pramlintide (Symlin) (monotherapy)	Alzheimer´s Disease; Mild Cognitive Impairment;	Plasma Aβ and t-tau changes. Plasma inflammatory changes. Metabolic changes in blood. All parameters determined 5, 30, 60, and 180 min after a Pramlintide challenge test.	Boston University	Recruiting	31 July 2024
2	NCT00691158 N/A/RTBPC	Pramlintide (monotherapy or combined with Metreleptin) or placebo	Obesity	Hypothalamic, brainstem and whole brain fMRI response measurements to different treatments.	Oregon Health and Science University	Active, not recruiting	16 December 2022
3	NCT00690235 IV RDBPC	Pramlintide (monotherapy) or placebo	Schizophrenia; Schizoaffective Disorder; Diabetes; Weight Gain	Weight loss from Pramlintide in Olanzapine- or Clozapine-induced weight gain patients with schizophrenia	University of Texas Southwestern Medical Center	Completed	16 November 2018
4	NCT00189514 II/RDBPC	Pramlintide acetate (monotherapy) or placebo	Obesity	Long-term effects of Pramlintide on body weight in obese subjects. Long-term safety and tolerability of the drug in obese patients.	AstraZeneca	Completed	11 June 2015
5	NCT00444561 II/RDBPC	Pramlintide acetate (AC137, monotherapy) or placebo	Overweight; Obesity	Pharmacodynamics of Pramlintide in obese subjects.	AstraZeneca	Completed	11 June 2015
6	NCT00673387 II/RDBPC	Pramlintide acetate (Smylin) or placebo *versus* Metreleptin or placebo	Overweight; Obesity	Measurement of the mean percentage of change in body weight from baseline to week 28 of the study.	AstraZeneca	Completed	15 April 2015
7	NCT00819234 II/RTBPC	Pramlintide (combined with Metreleptin) or placebo	Obesity	Measurement of the mean percentage of change in body weight from the baseline of the original study (NCT00673387) at week 52 in the extension study.	AstraZeneca	Completed	15 April 2015
8	NCT01235741 II/RDBPC	Pramlintide (combined with Metreleptin) or placebo	Obesity	Effect of 16 weeks of treatment on body weight in subjects with obesity. Studies of safety and tolerability of the drug combination.	AstraZeneca	Terminated	15 April 2015
9	NCT0039292 IIa/RDBPC	Pramlintide acetate (Smylin) or placebo; Metreleptin or placebo	Overweight; Obesity	Effect of 16 weeks of treatment on body weight in participants with overweight and/or obesity.	AstraZeneca	Completed	14 April 2015
10	NCT00112021 IIb/RDBPC	Pramlintide acetate (monotherapy) or placebo	Obesity	Effect of 16 weeks of treatment on body weight in subjects with obesity. Studies of safety and tolerability of Pramlintide in participants with obesity.	AstraZeneca	Completed	10 April 2015
11	NCT00402077 II/RSBPC	Pramlintide acetate (Symlin) compared to Sibutramine, Phentermine or placebo	Overweight; Obesity	Determination of treatment-related adverse events during 24-week treatment period. Change in body weight from baseline to week 12 of treatment.	AstraZeneca	Completed	6 March 2015
	Cagrilintide						
12	NCT06131372 II/RQBPC	CagriSema or placebo *versus* active comparators [Cagrilintide or Semaglutide, monotherapies] or placebo	Chronic Kidney Disease; Type 2 Diabetes; Obesity	Change in urinary albumin-to-creatinine ratio (UACR) from baseline (week 0) to end of treatment (week 26).	Novo Nordisk A/S	Recruiting	26 August 2024
13	NCT06388187 III/RQBPC	CagriSema (two different doses) or placebo (two diferent doses)	Obesity	Body weight changes from baseline (week 0) to end of treatment (week 68). Achievement of ≥5% weight reduction from baseline to the end of the treatmen.	Novo Nordisk A/S	Recruiting	20 August 2024
14	NCT06131437 III/ROL	CagriSema versus active comparator (Tirzepatide)	Obesity	Change in body weight from baseline (week 0) to end of treatment (week 72).	Novo Nordisk A/S	Active, not recruiting	20 August 2024
15	NCT05567796 (REDEFINE 1) III/RQBPC	CagriSema or placebo *versus* active comparators [Cagrilintide or Semaglutide, monotherapies]	Obesity	Body weight changes from baseline (week 0) to end of treatment (week 68). Achievement of ≥5% weight reduction from baseline to the end of the treatment.	Novo Nordisk A/S	Active, not recruiting	20 August 2024
16	NCT05813925 III/RQBPC	CagriSema *versus* active comparator (Semaglutide)	Overweight; Obesity	Body weight changes from baseline (week 0) to end of treatment (week 68).	Novo Nordisk A/S	Recruiting	6 August 2024
17	NCT05394519 (REDEFINE 2) III/RQBPC	CagriSema or placebo	Overweight; Obesity; Type 2 Diabetes Mellitus	Body weight changes from baseline (week 0) to end of treatment (week 68). Achievement of ≥5% weight reduction from baseline to the end of the treatment.	Novo Nordisk A/S	Active, not recruiting	17 July 2024
18	NCT05996848 (REDEFINE 6) III/RDBPC	CagriSema *versus* active comparator [Semaglutide, monotherapy] or placebo	Overweight; Obesity	Body weight changes from baseline (week 0) to end of treatment (week 68). Achievement of ≥5% weight reduction from baseline to the end of the treatment.	Novo Nordisk A/S	Recruiting	1 July 2024
19	NCT05804162 I/RQBPC	Cagrilintide or placebo *versus* Moxifloxacin and placebo	Obesity	Change from baseline in Fredericia’s heart rate-corrected QT interval (QTcF).	Novo Nordisk A/S	Completed	30 October 2023
20	NCT04940078 I/ROL	Cagrilintide (monotherapy) *versus* CagriSema (Cagrilintide+Semaglutide)	Overweight; Obesity	Pharmacokinetic characterization of treatments in the subject population.	Novo Nordisk A/S	Completed	20 February 2024
21	NCT06289504 I/ROL	CagriSema +Atorvastatin + Warfarin *versus* Semaglutide +Atorvastatin + Warfarin	Obesity	Pharmacokinetic characterization of treatments in the subject population.	Novo Nordisk A/S	Recruiting	14 May 2024
22	NCT06207877 I/RQBPC	CagriSema or placebo	Obesity	Changes in energy intake during ad libitum lunch, evening meal and snackbox from baseline to day 156 of the study.	Novo Nordisk A/S	Recruiting	15 March 2024
23	NCT06267092 I/RQBPC	CagriSema or placebo	Overweight; Obesity	Change in mean postprandial appetite score based on visual analog scale (VAS) from baseline to week 24 of the study.	Novo Nordisk A/S	Recruiting	28 February 2024
	Davalintide (AC2307)						
24	NCT00785408 II/RDBPC	AC2307 or placebo	Overweight; Obesity	Changes in body weight from baseline to week 24 of the study. Studies of safety and tolerability.	AstraZeneca	Completed	19 January 2015
	GUBamy (GUB014295)						
25	NCT06144684 I/RDBPC	GUB014295 (GUBamy) or placebo	Healthy Volunteers; Overweight	Adverse events (AE) incidence. Changes in vital signs. Safety laboratory parameters. All safety parameters determined from baseline (day 0) to the end of the trial’s part 1 (day 29), part 2A (day 64) and part 2B (day 106).	Gubra A/S	Recruiting	21 August 2024
	Celastrol						
26	NCT05494112 N/A/Safety OL	Celastrol (dietary supplement)	Healthy Volunteers	Effect of Celastrol in on liver function.	Legend Labz, Inc.	Unknown status	9 September 2022

Abbreviations: OL, open label; RDPBC, randomized double-blind placebo-controlled; ROL, randomized open-label; RQPBC, randomized quadruple-blind placebo-controlled; RTPBC, randomized triple-blind placebo-controlled.

## Data Availability

The original data presented in Table 1 of this work are openly available in: https://clinicaltrials.gov/ (accessed on 16 October 2024).

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
