# Peer review of "Hippocampal Leptin Resistance and Cognitive Decline: Mechanisms, Therapeutic Strategies and Clinical Implications"

_biomedicines, 2024, doi:10.3390/biomedicines12112422_

Round 1

Reviewer 1 Report

Comments and Suggestions for Authors

The present article by Valladolid-Acebes provides a comprehensive review of leptin resistance and its impact on hippocampal function, focusing on long-term potentiation (LTP) and long-term depression (LTD). The review includes discussions on high-fat diets and potential therapeutic strategies, such as leptin sensitizers, caloric restriction, and exercise, adding an additional value to address cognitive impairment.
The article is very well-written, comprehensive, and provides a clear understanding of the topic. The flow of the article is coherent and is presented in a clear and concise manner.

I only have a few minor suggestions to further enhance the clarity and precision of the manuscript:

- In line 315: A reference is needed: “The potential irreversibility of diet-induced hippocampal damage, due to leptin resistance…”, since there is no previous discussion about the reversibility or not (plasticity) of this process.
- While the article focuses on learning and memory deficits, it might benefit from a broader discussion of other cognitive domains affected by leptin resistance, such as attention or executive function.
- The review could consider briefly discussing how leptin resistance and associated cognitive impairments could contribute to comorbid conditions, such as Alzheimer's disease, as there is some evidence linking metabolic dysfunction to this disorder.
- Unless required by the journal, I would add the conclusion in a separate independent section
- Line 146: The In the hypothalamus (please correct)
- Although is already mostly addressed in some cases, I would recommend to clearly distinguish when the findings discussed are derived from studies in rodents. This is important to avoid generalizations and too prevent confusions. For example, in line 229: “Direct leptin administration in the dentate gyrus or the CA1 region of the hippocampus enhance LTP and spatial memory [108,109]” would apply to rodents only.

Author Response

Reviewer 1 is sincerely acknowledged for the thorough and thoughtful feedback on this manuscript. The valuable suggestions made in the comments provided have significantly contributed to improving the clarity and precision of the manuscript, and I appreciate the effort taken to enhance its overall quality. Please, find below a point-by-point addressing of the questions raised:

Comment 1: In line 315: A reference is needed: “The potential irreversibility of diet-induced hippocampal damage, due to leptin resistance…”, since there is no previous discussion about the reversibility or not (plasticity) of this process.

Response 1: Reference 151 of the latest version of the manuscript has been added to this statement, which has been rephrased for more clarity as follows:

The potential irreversibility of diet obesity-induced hippocampal damage, due to leptin resistance, highlights the urgency of early interventions to prevent cognitive impairments and to address the addictive nature of food that undermines efforts to control intake, contributing to the perpetuation of obesity [151].

Ref. 151: Gómez-Apo, E.; Mondragón-Maya, A.; Ferrari-Díaz, M.; Silva-Pereyra, J. Structural Brain Changes Associated with Overweight and Obesity. J Obes 2021, 2021:6613385.

Comment 2: While the article focuses on learning and memory deficits, it might benefit from a broader discussion of other cognitive domains affected by leptin resistance, such as attention or executive function.

Response 2: To address this question, a new section has been generated in lines 326-349 of the latest version of this manuscript, entitled “7. Implications of hippocampal leptin resistance for Executive Function and Alzheimer's Disease”. In the new section 7, the manuscript benefit from the reviewer´s comment by providing a broader discussion on the deleterious effects of hippocampal leptin resistance in executive function and attention. Additionally, these points have been further connected to Comment 3 of Reviewer 1 in section 7. Finally, the point raised by the reviewer has been included in the main conclusions of the current version of the manuscript (lines 490-496).

Comment 3: The review could consider briefly discussing how leptin resistance and associated cognitive impairments could contribute to comorbid conditions, such as Alzheimer's disease, as there is some evidence linking metabolic dysfunction to this disorder.

Response 3: As mentioned in Answer 2, the new section 7 (lines 326-349) included in the main text of the current version of this work clarifies how leptin resistance and associated cognitive impairments in obesity are, as the reviewer very well points out, linked to neurodegenerative disorders, such as Alzheimer's disease. Furthermore, this linkage has been also included in the main conclusions of the new version of the current work (lines 490-496).

  1. Implications of hippocampal leptin resistance for Executive Function and Alzheimer's Disease

In rodent models and humans, obesity-induced leptin resistance in the hippocampus has significant implications for executive functions, including decision-making, impulse con-trol and attention regulation, which are vital for maintaining cognitive control over food intake [152-157]. When leptin resistance occurs in the hippocampus, these cognitive pro-cesses are impaired, leading to weakened appetite regulation and reinforcing maladaptive eating behaviours [152-157]. This dysfunction can diminish working memory and execu-tive function, making it challenging for individuals to prioritize long-term goals over im-mediate food-related impulses [152-157]. These derangements become even more pro-nounced in neurodegenerative conditions such as Alzheimer's disease [85,116,158,159]. In patients with Alzheimer's disease, leptin signalling in the hippocampus is decreased and leptin localization is shifted being more abundant in reactive astrocytes and less in neu-rons [160]. These deranged hippocampal leptin signalling can very well be a contributing factor to neuronal degeneration, linking the metabolic dysfunction directly to cognitive decline in Alzheimer's disease [160].

Both leptin resistance in obesity and disrupted leptin signalling in Alzheimer's disease share a common pathway of cognitive impairment, particularly affecting memory and executive function both in rodents and humans [85,116,158-160]. This relationship un-derscores the potential of targeting leptin pathways as a therapeutic strategy to mitigate cognitive decline and address the metabolic aspects of neurodegenerative diseases [85,116,158-160]. By understanding these mechanisms, we can better explore interven-tions that may enhance leptin sensitivity and improve cognitive outcomes in affected in-dividuals.  

  1. Conclusion

In conclusion, the emerging understanding of hippocampal leptin resistance reveals its profound impact on cognitive function [36,41-46,98,128-132,139-151], particularly in executive processes such as decision-making, impulse control and memory [152-157]. This dysregulation not only exacerbates the challenges of obesity but also contributes to the pathophysiology of neurodegenerative diseases like Alzheimer’s disease [85,116,158,159]. As leptin signaling is compromised in these conditions, the link between metabolic dysfunction and cognitive decline becomes increasingly apparent [160].

Comment 4: Unless required by the journal, I would add the conclusion in a separate independent section

Response 4: Thanks for raising this point. The conclusion has been added in a separate independent section, upon the reviewer´s request, which is in the current version section 10. This section has been expanded more deeply and significantly, as shown in lines 489-515.

  1. Conclusion

In conclusion, the emerging understanding of hippocampal leptin resistance reveals its profound impact on cognitive function [36,41-46,98,128-132,139-151], particularly in executive processes such as decision-making, impulse control and memory [152-157]. This dysregulation not only exacerbates the challenges of obesity but also contributes to the pathophysiology of neurodegenerative diseases like Alzheimer’s disease [85,116,158,159]. As leptin signaling is compromised in these conditions, the link between metabolic dysfunction and cognitive decline becomes increasingly apparent [160].

Addressing hippocampal leptin resistance presents a promising avenue for thera-peutic intervention [161-163]. Targeted strategies, including the development of leptin sensitizers and lifestyle modifications [47,48,173-186], may enhance cognitive outcomes while also mitigating the obesity epidemic [194]. These approaches hold the potential to restore synaptic plasticity and improve neurogenesis, critical components for maintaining cognitive (Table 1, NCT03560960).

However, the complexity of the role of leptin in the central nervous system needs further investigation into the mechanisms underlying its effects [4,6,23-27]. Unraveling these pathways could lead to innovative treatments that specifically enhance leptin sig-naling within the brain, thereby alleviating cognitive deficits without exacerbating pe-ripheral resistance [85,116,158-160].

Given the increasing prevalence of obesity and its associated cognitive impairments [194,195], early interventions targeting leptin resistance are essential. Ultimately, fostering an integrated understanding of the interplay between metabolism and cognition [160] will be crucial in developing effective strategies to improve brain health and quality of life for individuals affected by these conditions. The time is ripe for a multifaceted approach that bridges clinical research, pharmacotherapy and lifestyle interventions [47,48,173-186], to combat the deleterious effects of hippocampal leptin resistance and enhance cognitive re-silience.

Comment 5: Line 146: The In the hypothalamus (please correct)

Response 5: Thank you very much for raising this mistake which is located now in line 148. It has now been corrected accordingly:

The In the hypothalamus,

Comment 6: Although is already mostly addressed in some cases, I would recommend to clearly distinguish when the findings discussed are derived from studies in rodents. This is important to avoid generalizations and too prevent confusions. For example, in line 229: “Direct leptin administration in the dentate gyrus or the CA1 region of the hippocampus enhance LTP and spatial memory [108,109]” would apply to rodents only.

Response 6: Certainly, the reviewer´s comment helps out to avoid generalizations and the following changes have been made in the main text:

-Line 69: …learning and memory processes both in rodents and in humans [42-45 40]…

-Line 70: Emerging Preclinical evidence suggests that leptin resistance impairs hippocampal…

-Line 73: … leptin signaling in the mouse brain point to a broader impact on cognitive functions and… 

-Line 76: … beneficial for cognitive deficits in both rodents and humans, albeit complicated by the…

-Line 222: … in the mouse liver [101,102]. In adipose tissue, leptin administration promotes lipolysis in…

-Line 223: … In rat skeletal muscle,…

-Line 225: Cardiac tissue from rats also benefits from leptin,…

-Line 229: … Preclinical Evidence suggests that leptin influences…

-Line 232: …or the CA1 region of the hippocampus enhance LTP and spatial memory in rodents…

-Line 296: … which in turn can exacerbate overeating and obesity in rodents and humans…

Reviewer 2 Report

Comments and Suggestions for Authors

The review article is very interesting and is devoted to the problem of leptin resistance in obesity and cognitive impairment. The review is very significant, well structured.

My comments:

1. The author analyzed a large number of literary sources (187 pieces). However, only 19% of sources from the last 5 years, this is not enough. The review needs to be updated with sources, especially for 2021-2024. I believe that it will not be difficult for the author to do this.

2. Table 1 is copied from source No. 53. The table is very overloaded and difficult to perceive. I suggest the author of the review to reduce the number of columns in it, modernize the table and thereby make his own authorial contribution to the analysis of these trials.

3. I do not like the conclusion (lines 425-429). I suggest the author to expand the conclusion more deeply and significantly.

Author Response

Reviewer 2 is sincerely thanked a thoughtful and constructive feedback on this work. The insights provided are invaluable in enhancing the quality of this manuscript. I appreciate the reviewer´s recognition of the importance and structure of this review and will address each suggestion in detail for greater depth and clarity. Please, find below a point-by-point addressing to the reviewer´s comments:

Comment 1. The author analyzed a large number of literary sources (187 pieces). However, only 19% of sources from the last 5 years, this is not enough. The review needs to be updated with sources, especially for 2021-2024. I believe that it will not be difficult for the author to do this.

Response 1: In the current version of the manuscript more recent literature has been incorporated and now more than 40% of the references correspond to sources from the last five years (concretely 79 out of 193 pieces). For more detail the following references have been added/substituted:

  1. Casado, M.E.; Collado-Pérez, R.; Frago, L.M.; Barrios, V. Recent Advances in the Knowledge of the Mechanisms of Leptin Physiology and Actions in Neurological and Metabolic Pathologies. Int J Mol Sci 2023, 24(2):1422.

  1. Boyle, C.A.; Kola, P.K.; Oraegbuna, C.S.; Lei, S. Leptin excites basolateral amygdala principal neurons and reduces food intake by LepRb-JAK2-PI3K-dependent depression of GIRK channels. J Cell Physiol 2024 239(2):e31117.

  1. Saxton, R.A.; Caveney, N.A.; Moya-Garzon, M.D.; Householder, K.D.; Rodriguez, G.E.; Burdsall, K.A.; Long, J.Z.; Garcia, K.C. Structural insights into the mechanism of leptin receptor activation. Nat Commun 2023, 14(1):1797.

Former references 26-26 have been removed and substituted by reference 29. Liu J, Lai F, Hou Y, Zheng R. Leptin signaling and leptin resistance. Med Rev 2022, 2(4):363-384

Former references 35-38 have been removed and substituted by reference 36. Madison, F.N.; Bingman, V.P.; Smulders, T.V.; Lattin, C.R. A bird's eye view of the hippocampus beyond space: Behavioral, neuroanatomical, and neuroendocrine perspectives. Horm Behav 2024 , 157:105451.

Former references 42-45 have been removed and substituted by reference 40. Bin Ibrahim, M.Z.; Benoy, A.; Sajikumar, S. Long-term plasticity in the hippocampus: maintaining within and 'tagging' between synapses. FEBS J 2022 , 289(8):2176-2201.

  1. Boyle, C.N.; Zheng, Y.; Lutz, T.A. Mediators of Amylin Action in Metabolic Control. J Clin Med 2022, 11(8):2207.

  1. Fan, X.; Qin, R.; Yuan, W.; Fan, J.S.; Huang, W.; Lin, Z. The solution structure of human leptin reveals a conformational plasticity important for receptor recognition. Structure 2024, 32(1):18-23.e2

  1. Picó, C.; Palou, M.; Pomar, C.A.; Rodríguez, A.M.; Palou, A. Leptin as a key regulator of the adipose organ. Rev Endocr Metab Disord 2022, 23(1):13-30

  1. Ansarin, A.; Mahdavi, A.M.; Javadivala, Z.; Shanehbandi, D.; Zarredar, H.; Ansarin, K. The cross-talk between leptin and circadian rhythm signaling proteins in physiological processes: a systematic review. Mol Biol Rep 2023, 50(12):10427-10443

  1. Harris, R.B.S. Phosphorylation of STAT3 in hypothalamic nuclei is stimulated by lower doses of leptin than are needed to inhibit food intake. Am J Physiol Endocrinol Metab 2021, 321(1):E190-E201

  1. Kommaddi, R.P.; Gowaikar, R.; Haseena, P.A.; Diwakar, L.; Singh, K.; Mondal, A. Akt activation ameliorates deficits in hippocampal-dependent memory and activity-dependent synaptic protein synthesis in an Alzheimer's disease mouse model. J Biol Chem 2024, 300(2):105619

  1. Mishra, P.; Narayanan, R. Stable continual learning through structured multiscale plasticity manifolds. Curr Opin Neurobiol 2021, 70:51-63

Former references 95-97 have been removed and substituted by reference 97. Ameroso, D.; Meng, A.; Chen, S.; Felsted, J.; Dulla, C.G.; Rios, M. Astrocytic BDNF signaling within the ventromedial hypothalamus regulates energy homeostasis. Nat Metab 2022, 4(5):627-643

  1. Kuhn, H.G.; Skau, S.; Nyberg, J. A lifetime perspective on risk factors for cognitive decline with a special focus on early events. Cereb Circ Cogn Behav 2024, 6:100217

  1. Kawai, T.; Autieri, M.V.; Scalia, R. Adipose tissue inflammation and metabolic dysfunction in obesity. Am J Physiol Cell Physiol 2021, 320(3):C375-C391

References 122, 139,141,143 have been removed

And References 151-160 have been added upon the request of Reviewer 1 for a new section discussing the implications of hippocampal leptin resistance for Executive Function and Alzheimer's Disease

Comment 2. Table 1 is copied from source No. 53. The table is very overloaded and difficult to perceive. I suggest the author of the review to reduce the number of columns in it, modernize the table and thereby make his own authorial contribution to the analysis of these trials.

Response 2: The number of columns has been reduced by removing the delivery methods, the eligible ages, the locations and by merging the phase and design of each trial with the column “NCT number”. Additionally, in the current cells containing the information of “NCT number, Phase and Design” of each trial have been hyperlinked to the full information provided by https://clinicaltrials.gov/, thus modernizing the table and making each trial more accessible for the readership to access the whole information currently published about each clinical trial. Finally, to make the table more visible and less overloaded, it has been divided into 4 landscaped pages.

Comment 3. I do not like the conclusion (lines 425-429). I suggest the author to expand the conclusion more deeply and significantly.

Response 3: Thanks for raising this point. The conclusion has been added in a separate independent section, upon the reviewer´s request, which is in the current version section 10. This section has been expanded more deeply and significantly, as shown in lines 489-515.

  1. Conclusion

In conclusion, the emerging understanding of hippocampal leptin resistance reveals its profound impact on cognitive function [36,41-46,98,128-132,139-151], particularly in executive processes such as decision-making, impulse control and memory [152-157]. This dysregulation not only exacerbates the challenges of obesity but also contributes to the pathophysiology of neurodegenerative diseases like Alzheimer’s disease [85,116,158,159]. As leptin signaling is compromised in these conditions, the link between metabolic dysfunction and cognitive decline becomes increasingly apparent [160].

Addressing hippocampal leptin resistance presents a promising avenue for thera-peutic intervention [161-163]. Targeted strategies, including the development of leptin sensitizers and lifestyle modifications [47,48,173-186], may enhance cognitive outcomes while also mitigating the obesity epidemic [194]. These approaches hold the potential to restore synaptic plasticity and improve neurogenesis, critical components for maintaining cognitive (Table 1, NCT03560960).

However, the complexity of the role of leptin in the central nervous system needs further investigation into the mechanisms underlying its effects [4,6,23-27]. Unraveling these pathways could lead to innovative treatments that specifically enhance leptin sig-naling within the brain, thereby alleviating cognitive deficits without exacerbating pe-ripheral resistance [85,116,158-160].

Given the increasing prevalence of obesity and its associated cognitive impairments [194,195], early interventions targeting leptin resistance are essential. Ultimately, fostering an integrated understanding of the interplay between metabolism and cognition [160] will be crucial in developing effective strategies to improve brain health and quality of life for individuals affected by these conditions. The time is ripe for a multifaceted approach that bridges clinical research, pharmacotherapy and lifestyle interventions [47,48,173-186], to combat the deleterious effects of hippocampal leptin resistance and enhance cognitive re-silience.

Round 2

Reviewer 2 Report

Comments and Suggestions for Authors

Dear Editor,

The authors have corrected the article according to all my comments. The review has become much better. I have no more comments.